# Software Quality: How Much Does It Matter?

Peter Kokol

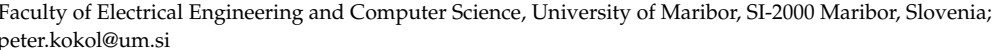

Faculty of Electrical Engineering and Computer Science, University of Maribor, SI-2000 Maribor, Slovenia; peter.kokol@um.si

**Abstract:** Interconnected computers and software systems have become an indispensable part of people's lives in the period of digital transformation. Consequently, software quality research is becoming more and more critical. There have been multiple attempts to synthesise knowledge gained in software quality research; however, they were focused mainly on single aspects of software quality and did not structure the knowledge holistically. To fill this gap, we harvested software quality publications indexed in the Scopus bibliographic database. We analysed them using synthetic content analysis which is a triangulation of bibliometrics and content analysis. The search resulted in 15,468 publications. The performance bibliometric analysis showed that the production of research publications relating to software quality is currently following an exponential growth trend and that the software quality research community is growing. The most productive country was the United States, followed by China. The synthetic content analysis revealed that the published knowledge could be structured into six themes, the most important being the themes regarding software quality improvement by enhancing software engineering, advanced software testing and improved defect and fault prediction with machine learning and data mining.

**Keywords:** software engineering; software quality; knowledge synthesis; bibliometrics; synthetic knowledge synthesis

## 1. Introduction

The digital transformation caused interconnected computers and software systems to become an indispensable part of our lives and the necessary toll for performing their daily personal and business obligations and activities [1]. To fulfil global user needs for storing, retrieving and processing information, knowledge and wisdom, those systems have to be supported by quality software, which should function correctly and reliably, be easy, safe and fit to use, test, reuse and maintain, and finally to conform to stakeholders' requirements. Therefore, software quality is not only one of the most important but also a multidimensional attribute of computer software [2–4].

Consequently, there have been multiple attempts to synthesise knowledge gained in software quality research; however, they were focused mainly on single aspects of software quality, such as measurement [5], human-software interaction [6], empirical analysis of Code smells and refactoring [7], design patterns [8]; quality models [9], human factors [10], testing [11] and quality prediction [12,13] or to specific development approaches such as agile [14].

Bibliometrics is another approach to synthesise knowledge [15]. Similarly to the above, the only two bibliometrics studies concerned with software quality were focused on specific aspects of software quality, one on defect prediction [16] and the other on code smells [17].

To close the gap regarding the lack of holistic knowledge synthesis studies of software quality research, we performed a performance bibliometric analysis and synthetic bibliometric mapping-based content analysis. We aimed to identify the most productive countries and institutions, most prolific source titles, publication production trends and hot topics, and structure the research content into themes and trends.

The choice to focus on software quality was motivated by the belief that software quality assurance is a crucial factor in software development for a myriad of stakeholders such as software developers, theoreticians, practitioners and of course, software users. The study can help them gain new insights into the topic, deepen their knowledge or inform them about the trends and essential themes in software quality research. To the best of our knowledge, a similar quantitative and qualitative bibliometric study that provides an overview of the current state and development of software quality research from its beginnings to the present has not been conducted so far. Therefore, we eliminate this shortcoming through the present study and fill in the current gap.

## 2. Materials and Methods

Knowledge synthesis is an approach to deal with the explosive growth in research literature production. Knowledge synthesis roots date back more than 120 years. However, they became more popular in the 1960s [18] and even more commonly used toward the end of the millennia with the emergence of the evidence-based paradigm [19,20]. Content analysis is one of the more popular knowledge synthesis methods used in qualitative and quantitative research. Its main advantages are that it is content-sensitive, highly flexible and can be used to analyse many types of data in an inductive or deductive manner [21].

To enable the knowledge synthesis of several thousand or even ten thousand publications, Kokol et al. [22] triangulated descriptive bibliometrics, text mining and bibliometric mapping and content analysis into *Synthetic knowledge synthesis*. In addition to semi-automatic analysis of large corpora, the combination of the above approaches enables one to combine quantitative and qualitative analysis of the content and production of research publications. The qualitative part of the synthetic knowledge synthesis was performed with the *Algorithm* presented below:

1. Harvest the research publications concerning software quality. The corpus of retrieved publications represents the content to analyse and the output of Step 1.
2. Identify codes in the corpus using text mining and cluster them into an author's keyword landscape with bibliometric mapping. Authors' keywords were selected as codes since they most concisely present the content of a publication. The author's keyword landscape is the output from Step 2.
3. Condense author keywords with similar meanings into codes for every single cluster and analyse the links between codes in individual clusters, and then map them into categories which form the output from Step 3.
4. Analyse categories and name each cluster with an appropriate theme. The list of themes is the final output of the qualitative analysis.

In Step 1, the publications were harvested from Scopus (Elsevier, Amsterdam, The Netherlands), the largest abstract and citation database of peer-reviewed literature. The corpus was formed on 14 April 2022, for the whole period covered by Scopus. We used the search string "quality software" OR "quality of software" OR "software quality" in information source titles, abstracts, and keywords. The reliability of the search was assessed using recall (fraction of the documents that are relevant to the query that is successfully retrieved) information retrieval functions using 20 important software quality publications and ten eminent authors (19 publications and all authors were retrieved). The publications and authors were selected based on the discussions with colleagues concerned with software engineering research. Using Scopus built-in functions, we exported the following metadata to a CSV formatted corpus file: publication titles, authors' affiliations' details, source title, publication type, publishing year, author keywords and funding data.

Bibliometric mapping in Step 2 has been performed using VOSViewer software version 1.6.16 (Leiden University, The Netherlands). VOSViewer uses text mining to recognise various text entities in publications such as terms, keywords and country names. Next, it employs the mapping technique, Visualisation of Similarities (VoS), based on the co-word analysis. Landscapes are displayed in various ways to present different aspects of the

research publications' content [23]. Steps 3 and 4 were performed using the traditional content analysis process [21].

## 3. Results and Discussion

### 3.1. Descriptive Bibliometrics

The search resulted in 15,468 publications. Among them were 9457 conference papers, 4845 journal articles, 401 conference reviews, 294 book chapters, 283 reviews, 74 editorials, 63 books, 31 short papers and 20 other types of publications.

The first paper indexed in Scopus was a conference paper titled Teaching software development in a studio environment published in 1954 [24]. After that (Figure 1), the production was low, not exceeding five publications per year till 1978 when the linear growth trend began, followed by exponential growth starting in 2001. The peak productivity was reached in 2019 with 1082 publications. The trend of the number of conference proceedings and journal papers followed a similar pattern till 2001, then the trend split. The number of journal papers followed a steady exponential trend reaching 400 journal articles in 2020, while the number of conference papers had an explosive growth period from 2001 till 2007. After that, the number of conference papers remained steady till 2018 and peaked in 2019 with 701 papers. According to the above trends, we can identify three milestones in software quality research, namely: (1) The beginning of the seventies when the first software crisis emerged [25]; (2) the end of the seventies when software quality models and measurements started to gain in importance [26]; and finally (3) in the beginning of the new century with the advent of agile programming [27].

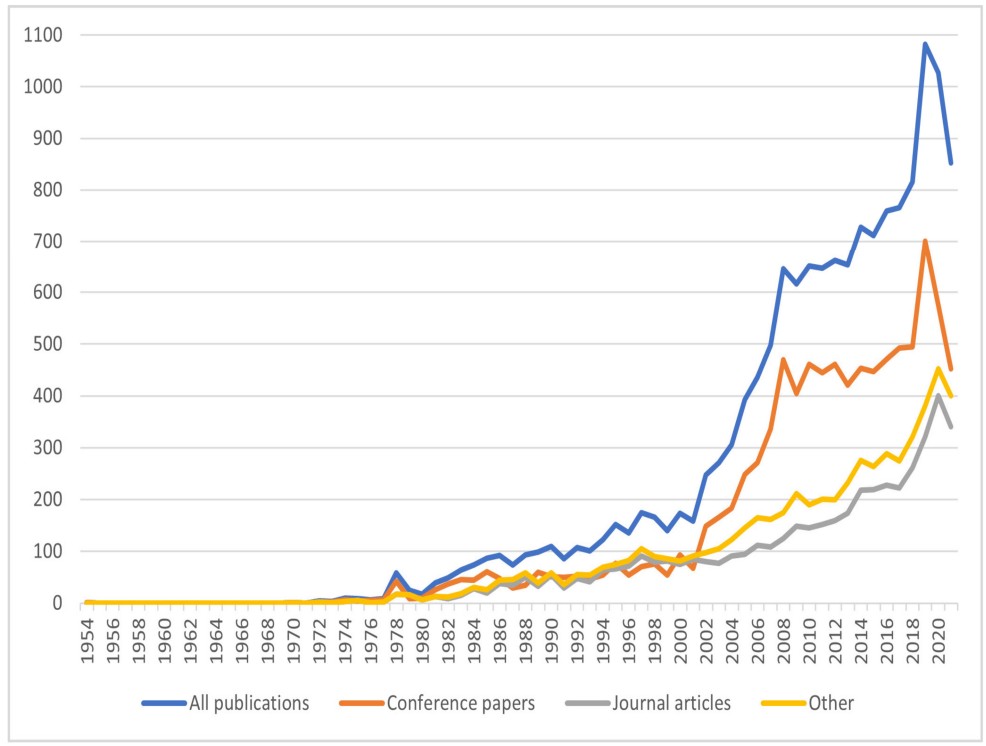

**Figure 1.** The dynamics of the software quality research expressed by the number of peer-reviewed publications.

Spatial Distribution and Productivity of Literature Production

The most prolific countries in software quality research according to the number of publications are presented in Table 1. Among them, India, China and Brazil belong among the 10 best countries to outsource software development [28]. The United States, Germany, India and the United Kingdom are the countries with the largest number of software engineers [29]. As expected, the most prolific institutions (Table 2) are primarily located in most prolific countries.

**Table 1.** Most productive countries.

| Country | Number of Publications |
|---|---|
| United States | 2917 |
| China | 1684 |
| India | 1436 |
| Germany | 961 |
| Canada | 734 |
| Brazil | 698 |
| United Kingdom | 626 |
| Italy | 510 |
| Spain | 428 |
| Japan | 427 |

**Table 2.** Most productive institutions.

| Institution | Number of Publications |
|---|---|
| Florida Atlantic University | 227 |
| Beihang University | 127 |
| Amity University | 99 |
| Peking University | 95 |
| Carnegie Mellon University | 87 |
| Universidade de São Paulo | 86 |
| Technical University of Munich | 83 |
| École de Technologie Supérieure | 80 |
| Chinese Academy of Sciences | 79 |
| Fraunhofer Institute for Experimental Software Engineering IESE | 75 |

The most prolific source titles (Table 3) are mainly conference and workshop proceedings. Most prolific journals are prestigious top software engineering journals with high SNIP 2020 impact factors. SNIP "measures contextual citation impact by weighting citations based on the total number of citations in a subject field, using Scopus data" [30].

**Table 3.** Most prolific journals.

| Source Title | Number of Publications | SNIP 2020 |
|---|---|---|
| Lecture Notes in Computer Science Including Subseries Lecture Notes in Artificial Intelligence And Lecture Notes In Bioinformatics | 737 | 0.628 |
| ACM International Conference Proceeding Series | 394 | 0.296 |
| Proceedings International Conference on Software Engineering | 324 | 1.68 |
| Communications In Computer and Information Science | 226 | 0.32 |
| Ceur Workshop Proceedings | 221 | 0.345 |
| Information And Software Technology | 214 | 2.389 |
| Software Quality Journal | 191 | 1.388 |
| Advances In Intelligent Systems and Computing | 190 | 0.428 |
| Journal Of Systems and Software | 186 | 2.16 |
| IEEE Software | 152 | 1.934 |

According to the number of publications where a funding agency was mentioned in the funding acknowledgement (Table 4), the most prolific funding agency is the National Natural Science Foundation of China ($n = 318$), followed by the US National Science Foundation ($n = 182$) and European Commission ($n = 132$). Based on funding acknowledgements analysis, only 21.5% of publications were sponsored, which is far less than for software engineering in general ($n = 32.3$%) [31]. That might indicate that software quality research is underfunded and did not yet reach the status of other software engineering disciplines.

Most productive funding institutions are located in China and Europe, and surprisingly not as expected by other bibliometrics performance attributes, in the United States.

**Table 4.** Most prolific funding agencies.

| Funding Agency | Number of Publications |
|---|---|
| National Natural Science Foundation of China | 318 |
| National Science Foundation (USA) | 182 |
| European Commission | 132 |
| Horizon 2020 Framework Programme (EU) | 97 |
| Conselho Nacional de Desenvolvimento Científico e Tecnológico (Brazil) | 89 |
| Natural Sciences and Engineering Research Council of Canada | 77 |
| Coordenação de Aperfeiçoamento de Pessoal de Nível Superior (Brazil) | 76 |
| National Key Research and Development Program of China | 66 |
| European Regional Development Fund | 65 |
| Japan Society for the Promotion of Science | 61 |

*3.2. Qualitative Synthetic Knowledge Synthesis*

3.2.1. Text Mining and Bibliometric Mapping

The 15,468 publications from the corpus were analysed using VOSviewer software (Step 2 of the algorithm). Text mining identified 19,084 author keywords. All keywords emerging in at least 40 papers (116 authors' keywords) were included in the bibliometric mapping analysis. The resulting author keywords landscape is shown in Figure 2.

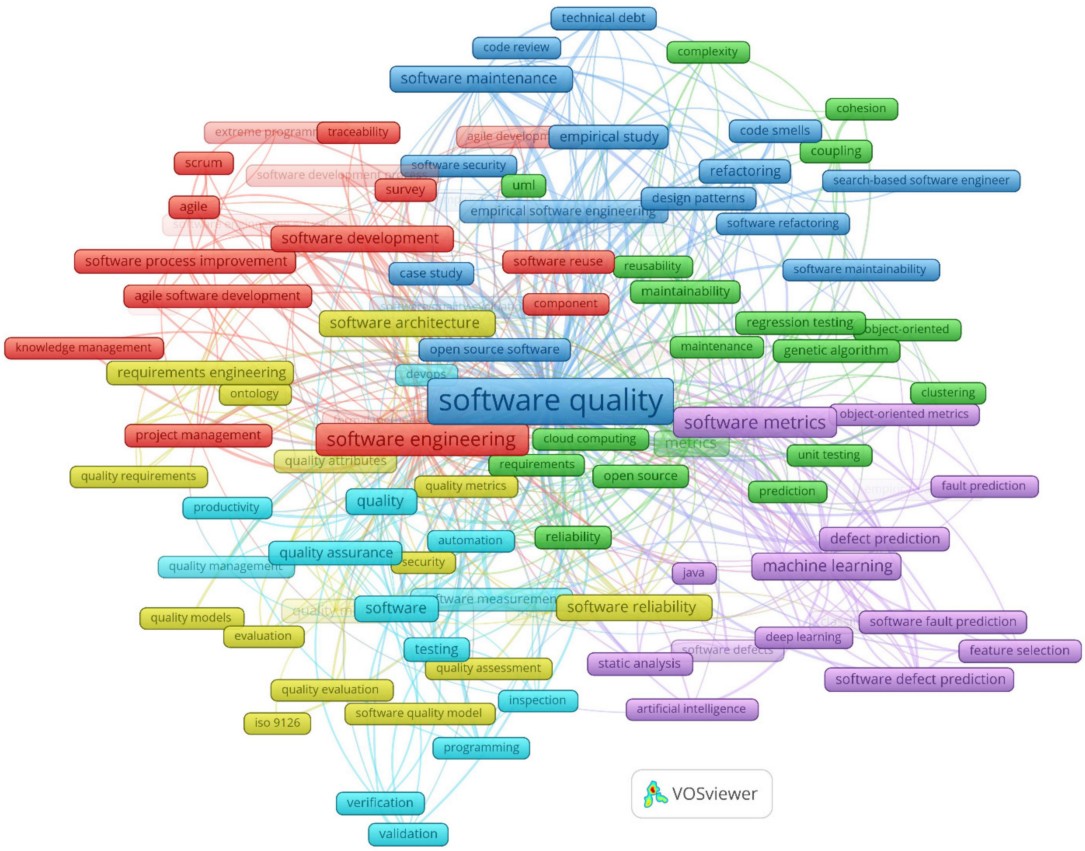

**Figure 2.** Authors' keywords cluster landscape.

3.2.2. Content Analysis

During the 3rd and 4th steps of the algorithm, author keywords were condensed into 34 codes. Those codes were next reduced into 16 categories. By analysing links between categories and codes, we finally identified six themes of software quality presented below and in Table 5:

- **Software process improvement**: The most impactful research regarding the number of citations was done around the beginning of the new millennia and was related to software process maturity [32,33] and the importance of top management leadership, management infrastructure and stakeholder participation [34]. Recent important research is still concerned with process maturity, but CMM(I) is combined with DevOps and agile approaches [35–37].
- **Metric-based software maintenance**: Most cited papers related to this theme were published in the period 1993–2010 and are concerned with object-oriented metrics to predict maintainability [38], metrics-based refactoring [39], predicting faults [40] and code readability metrics [41]. Recent impactful papers deal with technical debt [42], code smells and refactoring [43] and test smells [44].
- **Software evolution and refactoring:** The most influential research was published in the past decade and was concerned with the prediction of software evolution [45,46], automatic detection of bad smells which can trigger refactoring [47–49], the association of software defect and refactoring [50].
- **Quality assurance in initial phases of software development life cycles:** The impactful papers were published after the end of millennia. The research was mainly focused on how to assure software quality on the architectural level [51,52] and software requirements [53–55].
- **Search-based software engineering for defect prediction and classification:** The research on this theme has become important in the last 15 years. The research mainly used data mining and machine learning to predict software defects and failures using static codes and other software documents [56–58].
- **Software quality management and assurance with testing and inspections:** This theme is the most established, with influential papers starting to be published around 40 years ago [59–61]. Recent research is concerned with regression testing [11], predictive mutation testing [62], metamorphic testing [63] and modern code reviews [64].

*3.3. Hot Topics*

To identify hot topics, we extended the methodology developed by Kokol et al. [65] with the synthetic content analysis. We used this extended approach to compare the corpus of publications published in 2018 and 2019 with the corpus of publications published in 2020 and 2021. All the new categories and themes emerging in 2020/21, and the categories and themes emerging in 2018/19 mostly cited in 2020/21 were recognised as hot topics. In that manner, we identified:

- A new theme: Improving software development with the Integration of CMMI into agile approaches [66].
- New categories:
  - Natural language processing of software documents to elicit high-quality requirements [67].
  - Software quality attributes in agile environments [68].
  - Software architectures for the internet of things [69].
  - Software maintenance of blockchain-based software [70].
- Most cited categories in 2020/21:
  - Detecting code smells with genetic algorithms [71].
  - Use of Artificial intelligence in risk management [72].
  - Use multi-criteria decision-making in software quality modelling [73].

**Table 5.** Software quality research themes (numbers in parenthesis present the number of publications in which an author keyword appeared).

| Colour | Codes | Concepts/Categories | Themes |
|---|---|---|---|
| Red (26 keywords)) | Software engineering (641), Software quality assurance and management (330), Agile software development (319), Software development (276), Software process improvement (203), Software process (109), Software reuse (91), Project management (90) | Software quality assurance with project and knowledge management; software process improvement with agile approaches and CMMI; software reuse with production lines based on software quality metrics; | Software process improvement |
| Green (21 keywords) | Software testing (694), Metrics (578), Reliability (108); Maintainability (135), UML (80); Genetics algorithms (71) | Metrics-based software testing supported by genetic algorithms; using and predicting maintainability metrics like reusability, complexity, cohesion, and coupling; | Metric-based software maintenance |
| Blue (20 keywords) | Software quality (2685), Empirical studies in software engineering (281), Refactoring (226), Software maintenance (259), Technical depth and code smells (183), Software evolution (130),) | Mining software repositories to support empirical and search software engineering; software evolution with refactoring based on code smells; technical debts and code smells in association with software maintenance; | Software evolution and refactoring |
| Yellow (19 keywords) | Software architecture (253), Software reliability (233), Requirements engineering (226), Software quality models (214), Usability (108), Quality metrics and attributes (136), Quality assessment and evaluation (99) | Quality attributes of software requirements and architecture; quality attributes of software quality models and standards; general quality metrics like reliability; security, and usability; | Quality assurance in initial phases of software development life cycles |
| Viollet (16 keywords) | Software metrics (568), Machine learning and data mining (479); Fault and defect prediction (228) | Use of software metrics and data mining in defect prediction and classification; | Search-based software engineering for defect prediction and classification |
| Light blue (14 keywords) | Quality (221), Software (204); Quality management and assurance (164); Testing (148); | Testing and inspection, verification and validation; testing automation; programming productivity and quality | Software quality management and assurance with testing and inspections |

### 3.4. How Much Does the Software Quality Matter?

According to our synthesis, the volume, distribution and scope of quality research, as well as the research community and the research literature production, are growing, following the ever-growing importance of software in almost all human activities and recognising the potentially catastrophic consequences of quality defects, despite the fact that significant part of software quality research is still presented at conferences a robust, and well-researched body of evidence is forming but is not yet achieved in yet to be established core journals. Based on identified trends and best practices in overall software quality, substantial research in software quality exists but would need to be tai-

lored and extended to the specific requirements of software quality sub-disciplines and application domains.

### 3.5. Strengths and Limitations

The study's main strength is that it is the first holistic content analysis of software quality research. Another strength is that content analysis was performed using a novel synthetic knowledge synthesis approach. One possible limitation is that the analysis was limited to publications indexed in Scopus only; however, because Scopus covers the largest and most complete set of information titles, we believe that we analysed most of the important peer-reviewed publications.

### 4. Conclusions

Our bibliometric study showed that the production of research publications relating to software quality is currently following an exponential growth trend and that the software quality research community is growing. The synthetic content analysis revealed that the published knowledge could be structured into six themes, most important being the themes regarding software quality improvement by enhancing software engineering, advanced software testing and improved defect and fault prediction with machine learning and data mining. According to the analysis of the hot topics, it seems that future research will be directed into developing and using a whole spectre of new artificial intelligence tools (not just machine learning and data mining) and focusing on ensuring software quality in agile development paradigms.

The main contributions of or study are first, a holistic and better understanding of the software quality concept based on the triangulation of quantitative and qualitative analysis, which provides a landscape, performance and spatial attributes of the software quality research. Second, this study provides a paradigm for future studies in software development and digital transformation topics research.

**Funding:** This research was funded by ARRS, grant number J7-2605.

**Data Availability Statement:** Not applicable.

**Conflicts of Interest:** The author declares no conflict of interest.

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
