# Peer review of "Software Quality: How Much Does It Matter?"

_electronics, doi:10.3390/electronics11162485_

Round 1
Reviewer 1 Report
Dear Authors,
Review work is presented but there is a lack of clear outcomes. The algorithm is presented in Section 2 (line 75-86) but how these algorithm is implemented and also what is the outcomes at each stage is not clear. The algorithm in Section 2 has not linked or connection with Section 3- Results and Discussion. There should be clear linkage and connection. I found that it needs significant work before publication. Other comments are as follows:
1. Line number 92-95 - there is lack of clear explanation- how the author has selected 20 papers and 10 eminent authors?
2. Line 130-134- What is the basis for defining productive countries in software quality research and institutions? There is no reference or supporting arguments.
3. Line 139-141- written without any supporting arguments or data. What is SNIP 20202 in Table 3. It may be clear to you but not the reader- it is never defined. Number is also different format. check the table.
4. Reference [32] does not make any sense but it is self citation. The statement (at line no 143-145) - how did you achieve?
5. Found too much self-citations- eight self citations in a single paper. It is acceptable?
6. There is no clear conclusion and findings from the review work.
Good luck.
Author Response
Review work is presented but there is a lack of clear outcomes. The algorithm is presented in Section 2 (line 75-86) but how these algorithm is implemented and also what is the outcomes at each stage is not clear.
The description of the implementation of the algorithm and outputs from each step were made more explicit
The algorithm in Section 2 has not linked or connection with Section 3- Results and Discussion. There should be clear linkage and connection.
The Results and discussion section was amended and restructured in the way that it is now linked with Section 2
Line number 92-95 - there is lack of clear explanation- how the author has selected 20 papers and 10 eminent authors?
The explanation was added
- Line 130-134- What is the basis for defining productive countries in software quality research and institutions? There is no reference or supporting arguments.
The explanation was rewritten
- Line 139-141- written without any supporting arguments or data. What is SNIP 20202 in Table 3. It may be clear to you but not the reader- it is never defined. Number is also different format. check the table.
The paragraph was rewritten, and SNIP defined
- Reference [32] does not make any sense but it is self citation. The statement (at line no 143-145) - how did you achieve?
The paragraph was rewritten. Reference 32 is necessary, namely the paper presents the funding patterns in software engineering in general.
- Found too much self-citations- eight self citations in a single paper. It is acceptable?
The number of self-citations was reduced
- There is no clear conclusion and findings from the review work.
A subchapter summarising the findings was added
Reviewer 2 Report
The paper discusses a literature analysis with respect to software quality.
The paper is good to read and to follow. However there are some weaknesses, which require to go into more details.
The title is misleading, no "signs" of digital transformation (DT) can be found in the paper. DT is not discussed and seems not relevant to the found research papers. DT as a term can be only found in the title, abstract, keywords, intro and conclusions. No discussion on DT and the relationship to SW Quality can be found.
Also with respect to the title, the research does not answer the question how much the quality matters. The research identifies aspect and trends in SW Quality research.
At the end of the introduction section it is common sense to guide the reader towards whats on the road ahead and give a brief outlook on the upcoming sections.
line 64: paradigm. [21], [22]. -- remove first period
To better distinguish work that has been published in journals from work in conferences, I suggest to use the term journal article, not just article (I know, that the term in bibtex).
Not clear is the distinction between reviews and conference reviews.
Figure one has a yellow part, which is not explained somewhere.
I also have doubts about the blue line/area: that should be in total 1266 publications and hence just a bit higher (13%) than the conference line/area (which is 9457 publications). If you stack the number of papers. Additionally in the text it is said that in 2019 there are in total 1027 papers. But the blue line in fig. 1 is close to 1100.
In general I find fig.1 confusing. It might be easier to read if you do not stack the individual figures. Of course you could add a total line.
Figure 2 is missing.
The discussion of Fig. 3 is to short. Several keywords are clustered there, which require an explanation. Java, Cloud Computing, Genetic Algorithms, are the ones which - at least taking a first look at fig.3 make no sense. please explain. Maybe you need a "other category" where not quality related things go like "java".
More specifically, what has UML or Genetic Algorithms to do with "Metric-based software maintenance".
With respect to the keyword "software quality": This should not be assigned to a single category. Its the overall topic and as such will be addressed in all categories. Assigning a keyword to a cluster also implies that that keywords is less relevant (or not relevant at all) to other clusters. Which in this case is not true.
Please expand the discussion here. This part is definitely to short and leaves the reader with many questions.
Additionally, you might describe in detail the process which made you derive the proposed themes.
The "categories" in the hot topics sub-section are single papers. At least you quote only one of them in each. Thats a very limited number of papers to support your argument that this will be upcoming hot research.
The conclusions needs to be reworked. I find it also way to short. Also reflect on the process, What did you learn?
"The main contributions ...which provides a landscape, performance and spatial attributes of the software quality research" ... I do not see a landscape, neither performance (besides a few meta-data related data).
what do you mean with "spatial attributes" in this context?
"secondly, this study provides a paradigm for future studies in software development and digital transformation topics research"
no.
Author Response
The paper is good to read and to follow. However there are some weaknesses, which require to go into more details. The title is misleading, no "signs" of digital transformation (DT) can be found in the paper. DT is not discussed and seems not relevant to the found research papers. DT as a term can be only found in the title, abstract, keywords, intro and conclusions. No discussion on DT and the relationship to SW Quality can be found.
I do agree with the reviewer the title was changed to “Software quality: How much does it matter?”
Also with respect to the title, the research does not answer the question how much the quality matters. The research identifies aspect and trends in SW Quality research.
A subchapter answering the question was added
line 64: paradigm. [21], [22]. -- remove first period
Corrected
To better distinguish work that has been published in journals from work in conferences, I suggest to use the term journal article, not just article (I know, that the term in bibtex).
Corrected according to the reviewer suggestion
Not clear is the distinction between reviews and conference reviews.
Conference reviews are not ordinary reviews but the description/commentary of what happened at the conference
Figure one has a yellow part, which is not explained somewhere. I also have doubts about the blue line/area: that should be in total 1266 publications and hence just a bit higher (13%) than the conference line/area (which is 9457 publications). If you stack the number of papers. Additionally in the text it is said that in 2019 there are in total 1027 papers. But the blue line in fig. 1 is close to 1100. In general I find fig.1 confusing. It might be easier to read if you do not stack the individual figures. Of course you could add a total line.
The text and Figure 2 were corrected as sugested
Figure 2 is missing.
The figure numbering was incorrect and was corrected
The discussion of Fig. 3 is to short. Several keywords are clustered there, which require an explanation. Java, Cloud Computing, Genetic Algorithms, are the ones which - at least taking a first look at fig.3 make no sense. please explain. Maybe you need a "other category" where not quality related things go like "java". More specifically, what has UML or Genetic Algorithms to do with "Metric-based software maintenance".
The clusters were generated automatically using VOSviewer, During the content analysis, the irrelevant keywords or words indirectly linked to software quality were either removed or condensed into code and categories. For example, UML is indirectly connected to quality because it presents requirements more formally and thus requirements can be metricised and optimised for maintenance. In the same manner, genetic algorithms can be used to optimise the maintenance process based on complexity metrics of software components
Reviewer 3 Report
The paper performs a survey on software quality papers published in Scopus. They consider more than 15000 papers, which is a large set. They provide insights of the research trends and analysis metrics mapped with different time periods. The "Materials and Methods" section needs a detailed description. Which clustering methods have been used? What are the 20 important software quality publications, and who are the ten eminent authors? How have you defined them?
There are some formatting issues. Inconsistent font and margin in the text and table. Need to fix that.
Author Response
There are some formatting issues. Inconsistent font and margin in the text and table. Need to fix that
The formatting issues have been fixed
Round 2
Reviewer 1 Report
Dear Author,
Happy with the response and good luck.
Regards,
Rajan